# Graph Matching Networks for Learning the Similarity of Graph Structured Objects

## Abstract

This paper addresses the challenging problem of retrieval and matching of graph structured objects, and makes two key contributions. First, we demonstrate how Graph Neural Networks (GNN), which have emerged as an effective model for various supervised prediction problems defined on structured data, can be trained to produce embedding of graphs in vector spaces that enables efficient similarity reasoning. Second, we propose a novel Graph Matching Network model that, given a pair of graphs as input, computes a similarity score between them by jointly reasoning on the pair through a new cross-graph attention-based matching mechanism. We demonstrate the effectiveness of our models on different domains including the challenging problem of control-flow-graph based function similarity search that plays an important role in the detection of vulnerabilities in software systems. The experimental analysis demonstrates that our models are not only able to exploit structure in the context of similarity learning but they can also outperform domain-specific baseline systems that have been carefully hand-engineered for these problems.

## 1 Introduction

Graphs are natural representations for encoding relational structures that are encountered in many domains. Expectedly, computations defined over graph structured data are employed in a wide variety of fields, from the analysis of molecules for computational biology and chemistry (Gilmer et al., 2017; Yan et al., 2005), to the analysis of knowledge graphs or graph structured parses for natural language understanding.

In the past few years graph neural networks (GNNs) have emerged as an effective class of models for learning representations of structured data and for solving various supervised prediction problems on graphs. Such models are invariant to permutations of graph elements by design and compute graph node representations through a propagation process which iteratively aggregates local structural information (Scarselli et al., 2009; Li et al., 2015; Gilmer et al., 2017). These node representations are then used directly for node classification, or pooled into a graph vector for graph classification. Problems beyond supervised classification or regression are relatively less well-studied for GNNs.

In this paper we study the problem of similarity learning for graph structured objects, which appears in many important real world applications, in particular similarity based retrieval in graph databases. One motivating application is the computer security problem of binary function similarity search, where given a binary which may or may not contain code with known vulnerabilities, we wish to check whether any control-flow-graph in this binary is sufficiently similar to a database of known-vulnerable functions. This helps identify vulnerable statically linked libraries in closed-source software, a recurring problem (CVE, 2010; 2018) for which no good solutions are currently available. Figure 1 shows one example from this application, where the binary functions are represented as control flow graphs annotated with assembly instructions. This similarity learning problem is very challenging as subtle differences can make two graphs be semantically very different, while graphs with different structures can still be similar. A successful model for this problem should therefore (1) exploit the graph structures, and (2) be able to reason about the similarity of graphs both from the graph structures as well as from learned semantics.

In order to solve the graph similarity learning problem, we investigate the use of GNNs in this context, explore how they can be used to embed graphs into a vector space, and learn this embedding model to make similar graphs close in the vector space, and dissimilar graphs far apart. One important property of this model is that, it maps each graph independently to an embedding vector, and then all the similarity computation happens in the vector space. Therefore, the embeddings of graphs in a large database can be precomputed and indexed, which enables efficient retrieval with fast nearest neighbor search data structures like k-d trees (Bentley, 1975) or locality sensitive hashing (Gionis et al., 1999).

We further propose an extension to GNNs which we call Graph Matching Networks (GMNs) for similarity learning. Instead of computing graph representations independently for each graph, the GMNs compute a similarity score through a cross-graph attention mechanism to associate nodes across graphs and identify differences. By making the graph representation computation dependent on the pair, this matching model is more powerful than the embedding model, providing a nice accuracy-computation trade-off.

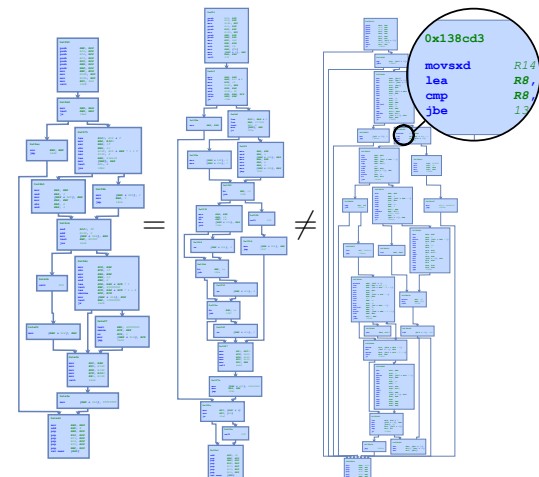

Figure 1: The binary function similarity learning problem. Checking whether two graphs are similar requires reasoning about both the structure as well as the semantics of the graphs. Here the left two control flow graphs correspond to the same function compiled with different compilers (and therefore similar), while the graph on the right corresponds to a different function.

We evaluate the proposed models and baselines on three tasks: a synthetic graph edit-distance learning task which captures structural similarity only, and two real world tasks - binary function similarity search and mesh retrieval, which require reasoning about both the structural and semantic similarity. On all tasks, the proposed approaches outperform established baselines and structure agnostic models; in more detailed ablation studies, we found that the Graph Matching Networks consistently outperform the graph embedding model and Siamese networks.

To summarize, the contributions of this paper are: (1) we demonstrate how GNNs can be used to produce graph embeddings for similarity learning; (2) we propose the new Graph Matching Networks that computes similarity through cross-graph attention-based matching; (3) empirically we show that the proposed graph similarity learning models achieve good performance across a range of applications, outperforming structure agnostic models and established hand-engineered baselines.

## 2 RELATED WORK

**Graph Neural Networks and Graph Representation Learning** The history of graph neural networks (GNNs) goes back to at least the early work by Gori et al. (2005) and Scarselli et al. (2009), who proposed to use a propagation process to learn node representations. These models have been further developed by incorporating modern deep learning components (Li et al., 2015; Veličković et al., 2017; Bruna et al., 2013). A separate line of work focuses on generalizing convolutions to graphs (Bruna et al., 2013; Bronstein et al., 2017). Popular graph convolutional networks also compute node updates by aggregating information in local neighborhoods (Kipf & Welling, 2016), making them the same family of models as GNNs. GNNs have been successfully used in many domains (Kipf & Welling, 2016; Veličković et al., 2017; Battaglia et al., 2016; 2018; Niepert et al., 2016; Duvenaud et al., 2015; Gilmer et al., 2017; Dai et al., 2017; Li et al., 2018; Wang et al., 2018a;b). Most of the previous work on GNNs focus on supervised prediction problems (with exceptions like (Dai et al., 2017; Li et al., 2018; Wang et al., 2018a)). The graph similarity learning problem we study in this paper and the new graph matching model can be good additions to this family of models.

**Graph Similarity Search and Graph Kernels** Graph similarity search has been studied extensively in database and data mining communities (Yan et al., 2005; Dijkman et al., 2009). The similarity

is typically defined by either exact matches (full-graph or sub-graph isomorphism) (Berretti et al., 2001; Shasha et al., 2002; Yan et al., 2004; Srinivasa & Kumar, 2003) or some measure of structural similarity, e.g. in terms of graph edit distances (Willett et al., 1998; Raymond et al., 2002). Most of the approaches proposed in this direction are not learning-based, and focus on efficiency.

Graph kernels are kernels on graphs designed to capture the graph similarity, and can be used in kernel methods for e.g. graph classification (Vishwanathan et al., 2010; Shervashidze et al., 2011). Popular graph kernels include those that measure the similarity between walks or paths on graphs (Borgwardt & Kriegel, 2005; Kashima et al., 2003; Vishwanathan et al., 2010), kernels based on limited-sized substructures (Horváth et al., 2004; Shervashidze et al., 2009) and kernels based on sub-tree structures (Shervashidze & Borgwardt, 2009; Shervashidze et al., 2011). Graph kernels are usually used in models that may have learned components, but the kernels themselves are hand-designed and motivated by graph theory. They can typically be formulated as first computing the feature vectors for each graph (the kernel embedding), and then take inner product between these vectors to compute the kernel value. One exception is (Yanardag & Vishwanathan, 2015) where the co-occurrence of graph elements (substructures, walks, etc.) are learned, but the basic elements are still hand-designed. Compared to these approaches, our graph neural network based similarity learning framework learns the similarity metric end-to-end.

**Distance Metric Learning**    Learning a distance metric between data points is the key focus of the area of metric learning. Most of the early work on metric learning assumes that the data already lies in a vector space, and only a linear metric matrix is learned to properly measure the distance in this space to group similar examples together and dissimilar examples to be far apart (Xing et al., 2003; Weinberger & Saul, 2009; Davis et al., 2007). More recently the ideas of distance metric learning and representation learning have been combined in applications like face verification, where deep convolutional neural networks are learned to map similar images to similar representation vectors (Chopra et al., 2005; Hu et al., 2014; Sun et al., 2014). In this paper, we focus on representation and similarity metric learning for graphs, and our graph matching model goes one step beyond the typical representation learning methods by modeling the cross-graph matchings.

**Siamese Networks**    Siamese networks (Bromley et al., 1994; Baldi & Chauvin, 1993) are a family of neural network models for visual similarity learning. These models typically consist of two networks with shared parameters applied to two input images independently to compute representations, a small network is then used to fuse these representations and compute a similarity score. They can be thought of as learning both the representations and the similarity metric. Siamese networks have achieved great success in many visual recognition and verification tasks (Bromley et al., 1994; Baldi & Chauvin, 1993; Koch et al., 2015; Bertinetto et al., 2016; Zagoruyko & Komodakis, 2015). In the experiments we adapt Siamese networks to handle graphs, but found our graph matching networks to be more powerful as they do cross-graph computations and therefore fuse information from both graphs early in the computation process. Independent of our work, recently (Shyam et al., 2017) proposed a cross-example attention model for visual similarity as an alternative to Siamese networks based on similar motivations and achieved good results.

## 3    Deep Graph Similarity Learning

Given two graphs $G_1 = (V_1, E_1)$ and $G_2 = (V_2, E_2)$, we want a model that produces the similarity score $s(G_1, G_2)$ between them. Each graph $G = (V, E)$ is represented as sets of nodes $V$ and edges $E$, optionally each node $i \in V$ can be associated with a feature vector $\mathbf{x}_i$, and each edge $(i, j) \in E$ associated with a feature vector $\mathbf{x}_{ij}$. These features can represent, e.g. type of a node, direction of an edge, etc. If a node or an edge does not have any associated features, we set the corresponding vector to a constant vector of 1s. We propose two models for graph similarity learning: a model based on standard GNNs for learning graph embeddings, and the new and more powerful GMNs. The two models are illustrated in Figure 2.

### 3.1    Graph Embedding Models

Graph embedding models embed each graph into a vector, and then use a similarity metric in that vector space to measure the similarity between graphs. Our GNN embedding model comprises 3 parts: (1) an encoder, (2) propagation layers, and (3) an aggregator.

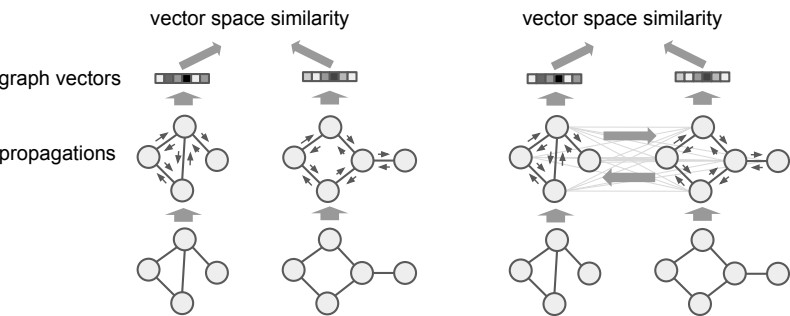

Figure 2: Illustration of the graph embedding (left) and matching models (right).

**Encoder** The encoder maps the node and edge features to initial node and edge vectors through separate MLPs:

$$\mathbf{h}_i^{(0)} = \mathrm{MLP}_{\mathrm{node}}(\mathbf{x}_i), \quad \forall i \in V \qquad \mathbf{e}_{ij} = \mathrm{MLP}_{\mathrm{edge}}(\mathbf{x}_{ij}), \quad \forall (i,j) \in E. \tag{1}$$

**Propagation Layers** A propagation layer maps a set of node representations $\{\mathbf{h}_i^{(t)}\}_{i \in V}$ to new node representations $\{\mathbf{h}_i^{(t+1)}\}_{i \in V}$, as the following:

$$\mathbf{m}_{j \to i} = f_{\mathrm{message}}(\mathbf{h}_i^{(t)}, \mathbf{h}_j^{(t)}, \mathbf{e}_{ij}), \qquad \mathbf{h}_i^{(t+1)} = f_{\mathrm{node}}\left(\mathbf{h}_i^{(t)}, \sum_{j:(j,i) \in E} \mathbf{m}_{j \to i}\right) \tag{2}$$

Here $f_{\mathrm{message}}$ is typically an MLP on the concatenated inputs, and $f_{\mathrm{node}}$ can be either an MLP or a recurrent neural network core, e.g. RNN, GRU or LSTM (Li et al., 2015). To aggregate the messages, we use a simple `sum` which may be alternatively replaced by other commutative operators such as `mean`, `max` or the attention-based weighted sum (Veličković et al., 2017). Through multiple layers of propagation, the representation for each node will accumulate information in its local neighborhood.

**Aggregator** After a certain number $T$ rounds of propagations, an aggregator takes the set of node representations $\{\mathbf{h}_i^{(T)}\}$ as input, and computes a graph level representation $\mathbf{h}_G = f_G(\{\mathbf{h}_i^{(T)}\})$, as

$$\mathbf{h}_G = \mathrm{MLP}_G\left(\sum_{i \in V} \sigma(\mathrm{MLP}_{\mathrm{gate}}(\mathbf{h}_i^{(T)})) \odot \mathrm{MLP}(\mathbf{h}_i^{(T)})\right), \tag{3}$$

which transforms node representations and then uses a weighted sum with gating vectors to aggregate across nodes. The weighted sum can focus only on the important nodes, it is more powerful than a simple sum and also works significantly better empirically.

After the graph representations $\mathbf{h}_{G_1}$ and $\mathbf{h}_{G_2}$ are computed for the pair $(G_1, G_2)$, we compute the similarity between them using a similarity metric in the vector space, for example the Euclidean, cosine or Hamming similarities.

Note that without the propagation layers (or with 0 propagation steps), this model becomes an instance of the Deep Set (Zaheer et al., 2017) or PointNet (Qi et al., 2017), which does computation on the individual nodes, and then pool the node representations into a representation for the whole graph. Such a model, however, ignores the structure and only treats the data as a set of independent nodes.

### 3.2 GRAPH MATCHING NETWORKS

Graph matching networks take a pair of graphs as input and compute a similarity score between them. Compared to the embedding models, these matching models compute the similarity score jointly on the pair, rather than first independently mapping each graph to a vector. Therefore these models are potentially stronger than the embedding models, at the cost of some extra computation efficiency.

We propose the following graph matching network, which changes the node update module in each propagation layer to take into account not only the aggregated messages on the edges for each graph as before, but also a cross-graph matching vector which measures how well a node in one graph can

be matched to one or more nodes in the other:

$$\mathbf{m}_{j \to i} = f_{\text{message}}(\mathbf{h}_i^{(t)}, \mathbf{h}_j^{(t)}, \mathbf{e}_{ij}), \quad \forall (i,j) \in E_1 \cup E_2 \tag{4}$$

$$\boldsymbol{\mu}_{j \to i} = f_{\text{match}}(\mathbf{h}_i^{(t)}, \mathbf{h}_j^{(t)}), \quad \forall i \in V_1, j \in V_2, \text{or } i \in V_2, j \in V_1 \tag{5}$$

$$\mathbf{h}_i^{(t+1)} = f_{\text{node}} \left( \mathbf{h}_i^{(t)}, \sum_j \mathbf{m}_{j \to i}, \sum_{j'} \boldsymbol{\mu}_{j' \to i} \right) \tag{6}$$

$$\mathbf{h}_{G_1} = f_G(\{\mathbf{h}_i^{(T)}\}_{i \in V_1}), \quad \mathbf{h}_{G_2} = f_G(\{\mathbf{h}_i^{(T)}\}_{i \in V_2}), \quad s = f_s(\mathbf{h}_{G_1}, \mathbf{h}_{G_2}). \tag{7}$$

Here $f_s$ is a standard vector space similarity between $\mathbf{h}_{G_1}$ and $\mathbf{h}_{G_2}$. $f_{\text{match}}$ is a function that communicates cross-graph information, which we propose to use an attention-based module:

$$a_{j \to i} = \frac{\exp(s_h(\mathbf{h}_i^{(t)}, \mathbf{h}_j^{(t)}))}{\sum_{j'} \exp(s_h(\mathbf{h}_i^{(t)}, \mathbf{h}_{j'}^{(t)}))}, \qquad \boldsymbol{\mu}_{j \to i} = a_{j \to i}(\mathbf{h}_i^{(t)} - \mathbf{h}_j^{(t)}) \quad \text{and therefore} \tag{8}$$

$$\sum_j \boldsymbol{\mu}_{j \to i} = \sum_j a_{j \to i}(\mathbf{h}_i^{(t)} - \mathbf{h}_j^{(t)}) = \mathbf{h}_i^{(t)} - \sum_j a_{j \to i}\mathbf{h}_j^{(t)}. \tag{9}$$

$s_h$ is again a vector space similarity metric, like Euclidean or cosine similarity, $a_{j \to i}$ are the attention weights, and $\sum_j \boldsymbol{\mu}_{j \to i}$ intuitively measures the difference between $\mathbf{h}_i^{(t)}$ and its closest neighbor in the other graph. Note that because of the normalization in $a_{j \to i}$, the function $f_{\text{match}}$ implicitly depends on the whole set of $\{\mathbf{h}_j^{(t)}\}$, which we omitted in Eq. 8 for a cleaner notation. Since attention weights are required for every pair of nodes across two graphs, this operation has a computation cost of $O(|V_1||V_2|)$, while for the GNN embedding model the cost for each round of propagation is $O(|V| + |E|)$. The extra power of the GMNs comes from utilizing the extra computation.

**Note** By construction, the attention module has a nice property that, when the two graphs can be perfectly matched, and when the attention weights are peaked at the exact match, we have $\sum_j \boldsymbol{\mu}_{j \to i} = \mathbf{0}$, which means the cross-graph communications will be reduced to zero vectors, and the two graphs will continue to compute identical representations in the next round of propagation. On the other hand, the differences across graphs will be captured in the cross-graph matching vector $\sum_j \boldsymbol{\mu}_{j \to i}$, which will be amplified through the propagation process, making the matching model more sensitive to these differences.

Compared to the graph embedding model, the matching model has the ability to change the representation of the graphs based on the other graph it is compared against. The model will adjust graph representations to make them become more different if they do not match.

## 3.3 LEARNING

The proposed graph similarity learning models can be trained on a set of example pairs or triplets. Pairwise training requires us to have a dataset of pairs labeled as positive (similar) or negative (dissimilar), while triplet training only needs relative similarity, i.e. whether $G_1$ is closer to $G_2$ or $G_3$. We describe the losses on pairs and triplets we used below, which are then optimized with gradient descent based algorithms.

When using Euclidean similarity, we use the following margin-based pairwise loss:

$$L_{\text{pair}} = \mathbb{E}_{(G_1, G_2, t)}[\max\{0, \gamma - t(1 - d(G_1, G_2))\}], \tag{10}$$

where $t \in \{-1, 1\}$ is the label for this pair, $\gamma > 0$ is a margin parameter, and $d(G_1, G_2) = \|\mathbf{h}_{G_1} - \mathbf{h}_{G_2}\|^2$ is the Euclidean distance. This loss encourages $d(G_1, G_2) < 1 - \gamma$ when the pair is similar ($t = 1$), and $d(G_1, G_2) > 1 + \gamma$ when $t = -1$. Given triplets where $G_1$ and $G_2$ are closer than $G_1$ and $G_3$, we optimize the following margin-based triplet loss:

$$L_{\text{triplet}} = \mathbb{E}_{(G_1, G_2, G_3)}[\max\{0, d(G_1, G_2) - d(G_1, G_3) + \gamma\}]. \tag{11}$$

This loss encourages $d(G_1, G_2)$ to be smaller than $d(G_1, G_3)$ by at least a margin $\gamma$.

For applications where it is necessary to search through a large database of graphs with low latency, it is beneficial to have the graph representation vectors be binary, i.e. $\mathbf{h}_G \in \{-1, 1\}^H$, so that

efficient nearest neighbor search algorithms (Gionis et al., 1999) may be applied. In such cases, we can minimize the Hamming distance of positive pairs and maximize it for negative pairs. With this restriction the graph vectors can no longer freely occupy the whole Euclidean space, but we gain the efficiency for fast retrieval and indexing. To achieve this we propose to pass the $\mathbf{h}_G$ vectors through a `tanh` transformation, and optimize the following pair and triplet losses:

$$L_{\text{pair}} = \mathbb{E}_{(G_1, G_2, t)}[(t - s(G_1, G_2))^2]/4, \quad \text{and} \tag{12}$$

$$L_{\text{triplet}} = \mathbb{E}_{(G_1, G_2, G_3)}[(s(G_1, G_2) - 1)^2 + (s(G_1, G_3) + 1)^2]/8, \tag{13}$$

where $s(G_1, G_2) = \frac{1}{H} \sum_{i=1}^{H} \tanh(h_{G_1 i}) \cdot \tanh(h_{G_2 i})$ is the approximate average Hamming similarity. Both losses are bounded in $[0, 1]$, and they push positive pairs to have Hamming similarity close to 1, and negative pairs to have similarity close to -1. We found these losses to be a bit more stable than margin based losses for Hamming similarity.

## 4 EXPERIMENTS

In this section, we evaluate the graph similarity learning (GSL) framework and the graph embedding (GNNs) and graph matching networks (GMNs) on three tasks and compare these models with other competing methods. Overall the empirical results demonstrate that the GMNs excel on graph similarity learning, consistently outperforming all other approaches.

### 4.1 LEARNING GRAPH EDIT DISTANCES

**Problem Background**    Graph edit distance between graphs $G_1$ and $G_2$ is defined as the minimum number of edit operations needed to transform $G_1$ to $G_2$. Typically the edit operations include add/remove/substitute nodes and edges. Graph edit distance is naturally a measure of similarity between graphs and has many applications in graph similarity search (Dijkman et al., 2009; Zeng et al., 2009; Gao et al., 2010). However computing the graph edit distance is NP-hard in general (Zeng et al., 2009), therefore approximations have to be used. Through this experiment we show that the GSL models can learn structural similarity between graphs on very challenging problems.

**Training Setup**    We generated training data by sampling random binomial graphs $G_1$ with $n$ nodes and edge probability $p$ (Erdös & Rényi, 1959), and then create positive example $G_2$ by randomly substituting $k_p$ edges from $G_1$ with new edges, and negative example $G_3$ by substituting $k_n$ edges from $G_1$, where $k_p < k_n$[1]. A model needs to predict a higher similarity score for positive pair $(G_1, G_2)$ than negative pair $(G_1, G_3)$. Throughout the experiments we fixed the dimensionality of node vectors to 32, and the dimensionality of graph vectors to 128 without further tuning. We also tried different number of propagation steps $T$ from 1 to 5, and observed consistently better performance with increasing $T$. The results reported in this section are all with $T = 5$ unless stated otherwise. More details are included in Appendix B.1.

**Baseline**    We compare our models with the popular Weisfeiler Lehman (WL) kernel (Shervashidze et al., 2011), which has been shown to be very competitive on graph classification tasks and the Weisfeiler Lehman algorithm behind this kernel is a strong method for checking graph isomorphism (edit distance of 0), a closely related task (Weisfeiler & Lehman, 1968; Shervashidze et al., 2011).

**Evaluation**    The performance of different models are evaluated using two metrics: (1) pair AUC - the area under the ROC curve for classifying pairs of graphs as similar or not on a fixed set of 1000 pairs and (2) triplet accuracy - the accuracy of correctly assigning higher similarity to the positive pair in a triplet than the negative pair on a fixed set of 1000 triplets.

**Results**    We trained and evaluated the GSL models on graphs of a few specific distributions with different $n, p$, with $k_p = 1$ and $k_n = 2$ fixed. The evaluation results are shown in Table 1. We can see that by learning on graphs of specific distributions, the GSL models are able to do better than generic baselines, and the GMNs consistently outperform the embedding model (GNNs).

For the GMNs, we can visualize the cross-graph attention to gain further insight into how it is working. Figure 3 shows two examples of this for a matching model trained with $n$ sampled from $[20, 50]$,

---

[1]Note that even though $G_2$ is created with $k_p$ edge substitutions from $G_1$, the actual edit-distance between $G_1$ and $G_2$ can be smaller than $k_p$ due to symmetry and isomorphism, same for $G_3$ and $k_n$. However the probability of such cases is typically low and decreases rapidly with increasing graph sizes.

| Graph Distribution | WL kernel | GNN | GMN |
|---|---|---|---|
| $n = 20, p = 0.2$ | 80.8 / 83.2 | 88.8 / 94.0 | **95.0 / 95.6** |
| $n = 20, p = 0.5$ | 74.5 / 78.0 | 92.1 / 93.4 | **96.6 / 98.0** |
| $n = 50, p = 0.2$ | 93.9 / **97.8** | 95.9 / 97.2 | **97.4** / 97.6 |
| $n = 50, p = 0.5$ | 82.3 / 89.0 | 88.5 / 91.0 | **93.8 / 92.6** |

Table 1: Comparing the graph embedding (GNN) and matching (GMN) models trained on graphs from different distributions with the baseline, measuring pair AUC / triplet accuracy ($\times 100$).

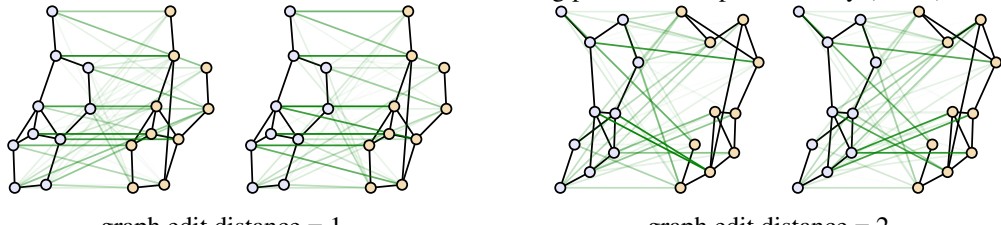

graph edit distance = 1                    graph edit distance = 2

Figure 3: Visualization of cross-graph attention for GMNs after 5 propagation layers. In each pair of graphs the left figure shows the attention from left graph to the right, the right figure shows the opposite.

tested on graphs of 10 nodes. The cross-graph attention weights are shown in green, with the scale of the weights shown as the transparency of the green edges. We can see that the attention weights can align nodes well when the two graphs match, and tend to focus on nodes with higher degrees when they don't. However the pattern is not as interpretable as in standard attention models.

More experiments on generalization capabilities of these models (train on small graphs, test on larger graphs, train on graphs with some $k_p, k_n$ combinations, test on others) are included in Appendix B.1.

### 4.2 CONTROL FLOW GRAPH BASED BINARY FUNCTION SIMILARITY SEARCH

**Problem Background**    Binary function similarity search is an important problem in computer security. The need to analyze and search through binaries emerges when we do not have access to the source code, for example when dealing with commercial or embedded software or suspicious executables. Combining a disassembler and a code analyzer, we can extract a control-flow graph (CFG) which contains all the information in a binary function in a structured format. See Figure 1 and Appendix B.2 for a few example CFGs. In a CFG, each node is a basic block of assembly instructions, and the edges between nodes represent the control flow, indicated by for example a jump or a return instruction used in branching, loops or function calls. In this section, we target the vulnerability search problem, where a piece of binary known to have some vulnerabilities is used as the query, and we search through a library to find similar binaries that may have the same vulnerabilities.[2] Accurate identification of similar vulnerabilities enables security engineers to quickly narrow down the search space and apply patches.

In the past the binary function similarity search problem has been tackled with classical graph theoretical matching algorithms (Eschweiler et al., 2016; Pewny et al., 2015), and Xu et al. (2017) and Feng et al. (2016) proposed to learn embeddings of CFGs and do similarity search in the embedding space. Xu et al. (2017) in particular proposed an embedding method based on graph neural networks, starting from some hand selected feature vectors for each node. Here we study further the performance of graph embedding and matching models, with pair and triplet training, different number of propagation steps, and learning node features from the assembly instructions.

**Training Setup and Baseline**    We train and evaluate our model on data generated by compiling the popular open source video processing software `ffmpeg` using different compilers `gcc` and `clang`, and different compiler optimization levels, which results in 7940 functions and roughly 8 CFGs per function. The average size of the CFGs is around 55 nodes per graph, with some larger graphs having up to a few thousand nodes (see Appendix B.2 for more detailed statistics). Different compiler optimization levels result in CFGs of very different sizes for the same function. We split the data and used 80% functions and the associated CFGs for training, 10% for validation and 10% for testing. The

---

[2]Note that our formulation is general and can also be applied to source code directly if they are available.

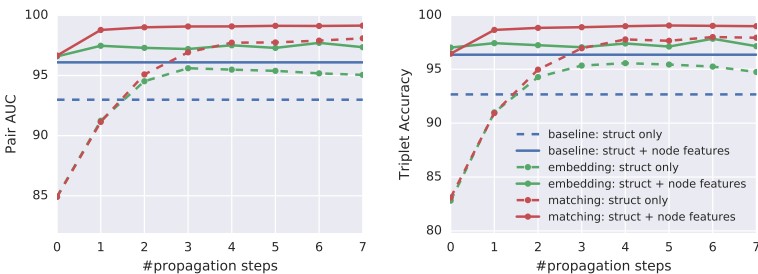

Figure 4: Performance ($\times 100$) of different models on the binary function similarity search task.

models were trained to learn a similarity metric on CFGs such that the CFGs for the same function have high similarity, and low similarity otherwise. Once trained, this similarity metric can be used to search through library of binaries and be invariant to compiler type and optimization levels.

We compare our graph embedding and matching models with Google's open source function similarity search tool (Dullien, 2018), which has been used to successfully find vulnerabilities in binaries in the past. This tool computes representations of CFGs through a hand-engineered graph hashing process which encodes the neighborhood structure of each node by hashing the degree sequence from a traversal of a 3-hop neighborhood, and also encodes the assembly instructions for each basic block by hashing the trigrams of assembly instruction types. These features are then combined by using a SimHash-style (Charikar, 2002) algorithm with learned weights to form a 128-dimensional binary code. An LSH-based search index is then used to perform approximate nearest neighbor search using hamming distance.

Following (Dullien, 2018), we also map the CFGs to 128-dimensional binary vectors, and use the Hamming similarity formulation described in Section 3 for training. We further studied two variants of the data, one that only uses the graph structure, and one that uses both the graph structure and the assembly instructions with learned node features. When assembly instructions are available, we embed each instruction type into a vector, and then sum up all the embedding vectors for instructions in a basic block as the initial representation vector (the $x_i$'s) for each node, these embeddings are learned jointly with the rest of the model.

**Results** Figure 4 shows the performance of different models with different number of propagation steps and in different data settings. We again evaluate the performance of these models on pair AUC and triplet accuracy on fixed sets of pairs and triplets from the test set. It is clear from results that: (1) the performance of both the graph embedding and matching models consistently go up with more propagation steps, and in particular significantly outperforming the structure agnostic model special case which uses 0 propagation steps; (2) the graph embedding model is consistently better than the baselines with enough propagation steps; and (3) graph matching models outperforms the embedding models across all settings and propagation steps. Additionally, we have tried the WL kernel on this task using only the graph structure, and it achieved 0.619 AUC and 24.5% triplet accuracy. This is not surprising as the WL kernel is not designed for solving this task, while our models learn the features useful for the task of interest, and can achieve better performance than generic similarity metrics.

### 4.3 MORE BASELINES AND ABLATION STUDIES

In this section, we carefully examine the effects of the design decisions we made in the GMN model and compare it against a few more alternatives. In particular, we evaluate the popular Graph Convolutional Network (GCN) model by Kipf & Welling (2016) as an alternative to our GNN model, and Siamese versions of the GNN/GCN embedding models. The GCN model replaces the message passing in Eq. 2 with graph convolutions, and the Siamese model predicts a distance value by concatenating two graph vectors and then pass through a 2 layer MLP. The comparison with Siamese networks can in particular show the importance of the cross-graph attention early on in the similarity computation process, as Siamese networks fuse the representations for 2 graphs only at the very end.

We focus on the function similarity search task, and also conduct experiments on an extra COIL-DEL mesh graph dataset (Riesen & Bunke, 2008), which contains 100 classes of mesh graphs corresponding to 100 types of objects. We treat graphs in the same class as similar, and used identical setup as the function similarity search task for training and evaluation.

| Model | Pair AUC | Triplet Acc |
|---|---|---|
| Baseline | 96.09 | 96.35 |
| GCN | 96.67 | 96.57 |
| Siamese-GCN | 97.54 | 97.51 |
| GNN | 97.71 | 97.83 |
| Siamese-GNN | 97.76 | 97.58 |
| GMN | **99.28** | **99.18** |

Function Similarity Search

| Model | Pair AUC | Triplet Acc |
|---|---|---|
| GCN | 94.80 | 94.95 |
| Siamese-GCN | 95.90 | 96.10 |
| GNN | 98.58 | 98.70 |
| Siamese-GNN | 98.76 | 98.55 |
| GMN | **98.97** | **98.80** |

COIL-DEL

Table 2: More results on the function similarity search task and the extra COIL-DEL dataset.

Table 2 summarizes the experiment results, which clearly show that: (1) the GNN embedding model is a competitive model (more powerful than the GCN model); (2) using Siamese network architecture to learn similarity on top of graph representations is better than using a prespecified similarity metric (Euclidean, Hamming etc.); (3) the GMNs outperform the Siamese models showing the importance of cross-graph information communication early in the computation process.

## 5 CONCLUSIONS AND DISCUSSION

In this paper we studied the problem of graph similarity learning using graph neural networks. Compared to standard prediction problems for graphs, similarity learning poses a unique set of challenges and potential benefits. For example, the graph embedding models can be learned through a classification setting when we do have a set of classes in the dataset, but formulating it as a similarity learning problem can handle cases where we have a very large number of classes and only very few examples for each class. The representations learned from the similarity learning setting can also easily generalize to data from classes unseen during training (zero-shot generalization).

We proposed the new graph matching networks as a stronger alternative to the graph embedding models. The added power for the graph matching models comes from the fact that they are not independently mapping each graph to an embedding, but rather doing comparisons at all levels across the pair of graphs, in addition to the embedding computation. The model can then learn to properly allocate capacity toward the embedding part or the matching part. The price to pay for this expressivity is the added computation cost in two aspects: (1) since each cross-graph matching step requires the computation of the full attention matrices, which requires at least $O(|V_1||V_2|)$ time, this may be expensive for large graphs; (2) the matching models operate on pairs, and cannot directly be used for indexing and searching through large graph databases. Therefore it is best to use the graph matching networks when we (1) only care about the similarity between individual pairs, or (2) use them in a retrieval setting together with a faster filtering model like the graph embedding model or standard graph similarity search methods, to narrow down the search to a smaller candidate set, and then use the more expensive matching model to rerank the candidates to improve precision.

Developing neural models for graph similarity learning is an important research direction with many applications. There are still many interesting challenges to resolve, for example to improve the efficiency of the matching models, study different matching architectures, adapt the GNN capacity to graphs of different sizes, and applying these models to new application domains. We hope our work can spur further research in this direction.

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

# A  EXTRA DETAILS ON MODEL ARCHITECTURES

In the propagation layers of the graph embedding and matching models, we used an MLP with one hidden layer as the $f_{\mathrm{message}}$ module, with a ReLU nonlinearity on the hidden layer. For node state vectors (the $\mathbf{h}_i^{(t)}$ vectors) of dimension $D$, the size of the hidden layer and the output is set to $2D$. We found it to be beneficial to initialize the weights of this $f_{\mathrm{message}}$ module to be small, which helps stablizing training. We used the standard Glorot initialization with an extra scaling factor of 0.1. When not using this small scaling factor, at the begining of training the message vectors when summed up can have huge scales, which is bad for learning.

One extra thing to note about the propagation layers is that we can make all the propagation layers share the same set of parameters, which can be useful if this is a suitable inductive bias to have.

We tried different $f_{\mathrm{node}}$ modules in both experiments, and found GRUs to generally work better than one-hidden layer MLPs, and all the results reported uses GRUs as $f_{\mathrm{node}}$, with the sum over edge messages $\sum_j \mathbf{m}_{j \to i}$ treated as the input to the GRU for the embedding model, and the concatenation of $\sum_j \mathbf{m}_{j \to i}$ and $\sum_{j'} \boldsymbol{\mu}_{j' \to i}$ as the input to the GRU for the matching model.

In the aggregator module, we used a single linear layer for the node transformation MLP and the gating $\mathrm{MLP}_{\mathrm{gate}}$ in Eq.3. The output of this linear layer has a dimensionality the same as the required dimensionality for the graph vectors. $\sigma(x) = \frac{1}{1+e^{-x}}$ is the logistic sigmoid function, and $\odot$ is the element-wise product. After the weighted sum, another MLP with one hidden layers is used to further transform the graph vector. The hidden layer has the same size as the output, with a ReLU nonlinearity.

For the matching model, the attention weights are computed as

$$a_{j \to i} = \frac{\exp(s_h(\mathbf{h}_i^{(t)}, \mathbf{h}_j^{(t)}))}{\sum_{j'} \exp(s_h(\mathbf{h}_i^{(t)}, \mathbf{h}_{j'}^{(t)}))}. \tag{14}$$

We have tried the Euclidean similarity $s_h(\mathbf{h}_i, \mathbf{h}_j) = -\|\mathbf{h}_i - \mathbf{h}_j\|^2$ for $s_h$, as well as the dot-product similarity $s_h(\mathbf{h}_i, \mathbf{h}_j) = \mathbf{h}_i^\top \mathbf{h}_j$, and they perform similarly without significant difference.

# B  EXTRA EXPERIMENT DETAILS

We fixed the node state vector dimensionality to 32, and graph vector dimensionality to 128 throughout both the graph edit distance learning and binary function similarity search tasks. We tuned this initially on the function similarity search task, which clearly performs better than smaller models. Increasing the model size however leads to overfitting for that task. We directly used the same setting for the edit distance learning task without further tuning. Using larger models there should further improve model performance.

## B.1  LEARNING GRAPH EDIT DISTANCES

In this task the nodes and edges have no extra features associated with them, we therefore initialized the $\mathbf{x}_i$ and $\mathbf{x}_{ij}$ vectors as vectors of 1s, and the encoder MLP in Eq.1 is simply a linear layer for the nodes and an identity mapping for the edges.

We searched through the following hyperparameters: (1) triplet vs pair training; (2) number of propagation layers; (3) share parameters on different propagation layers or not. Learning rate is fixed at 0.001 for all runs and we used the Adam optimizer Kingma & Ba (2014). Overall we found: (1) triplet and pair training performs similarly, with pair training slightly better, (2) using more propagation layers consistently helps, and increasing the number of propagation layers $T$ beyond 5 may help even more, (3) sharing parameters is useful for performance more often than not.

Intuitively, the baseline WL kernel starts by labeling each node by its degree, and then iteratively updates a node's representation as the histogram of neighbor node patterns, which is effectively also a graph propagation process. The kernel value is then computed as a dot product of graph representation vectors, which is the histogram of different node representations. When using the kernel with $T$ iterations of computation, a pair of graphs of size $|V|$ can have as large as a $2|V|T$

| Eval Graphs | WL kernel | GNN | GMN |
|---|---|---|---|
| $n = 100, p = 0.2$ | 98.5 / 99.4 | 96.6 / 96.8 | 96.8 / 97.7 |
| $n = 100, p = 0.5$ | 86.7 / 97.0 | 79.8 / 81.4 | 83.1 / 83.6 |
| $n = 200, p = 0.2$ | 99.9 / 100.0 | 88.7 / 88.5 | 89.4 / 90.0 |
| $n = 200, p = 0.5$ | 93.5 / 99.2 | 72.0 / 72.3 | 68.3 / 70.1 |

Table 3: Generalization performance on large graphs for the GSL models trained on small graphs with $20 \leq n \leq 50$ and $0.2 \leq p \leq 0.5$.

dimensional representation vector for each graph, and these sets of effective 'feature' types are different for different pairs of graphs as the node patterns can be very different. This is an advantage for WL kernel over our models as we used a fixed sized graph vector regardless of the graph size. We evaluate WL kernel for $T$ up to 5 and report results for the best $T$ on the evaluation set.

In addition to the experiments presented in the main paper, we have also tested the generalization capabilities of the proposed models, and we present the extra results in the following.

**Train on small graphs, generalize to large graphs.** In this experiment, we trained the GSL models on graphs with $n$ sampled uniformly from 20 to 50, and $p$ sampled from range $[0.2, 0.5]$ to cover more variability in graph sizes and edge density for better generalization, and we again fix $k_p = 1, k_n = 2$. For evaluation, we tested the best embedding models and matching models on graphs with $n = 100, 200$ and $p = 0.2, 0.5$, with results shown in Table 3. We can see that for this task the GSL models trained on small graphs can generalize to larger graphs than they are trained on. The performance falls off a bit on much larger graphs with much more nodes and edges. This is also partially caused by the fact that we are using a fixed sized graph vector throughout the experiments , but the WL kernel on the other hand has much more effective 'features' to use for computing similarity. On the other hand, as shown before, when trained on graphs from distributions we care about, the GSL models can adapt and perform much better.

**Train on some $k_p, k_n$ combinations, test on other combinations.** We have also tested the model trained on graphs with $n \in [20, 50]$, $p \in [0.2, 0.5]$, $k_p = 1, k_n = 2$, on graphs with different $k_p$ and $k_n$ combinations. In particular, when evaluated on $k_p = 1, k_n = 4$, the models perform much better than on $k_p = 1, k_n = 2$, reaching 1.0 AUC and 100% triplet accuracy easily, as this is considerably simpler than the $k_p = 1, k_n = 2$ setting. When evaluated on graphs with $k_p = 2, k_n = 3$, the performance is workse than $k_p = 1, k_n = 2$ as this is a harder setting.

In addition, we have also tried training on the more difficult setting $k_p = 2, k_n = 3$, and evaluate the models on graphs with $k_p = 1, k_n = 2$ and $n \in [20, 50], p \in [0.2, 0.5]$. The performance of the models on these graphs are actually be better than the models trained on this setting of $k_p = 1, k_n = 2$, which is surprising and clearly demonstrates the value of good training data. However, in terms of generalizing to larger graphs models trained on $k_p = 2, k_n = 3$ does not have any significant advantages.

## B.2 BINARY FUNCTION SIMILARITY SEARCH

In this task the edges have no extra features so we initialize them to constant vectors of 1s, and the encoder MLP for the edges is again just an identity mapping. When using the CFG graph structure only, the nodes are also initialized to constant vectors of 1s, and the encoder MLP is a linear layer. In the case when using assembly instructions, we have a list of assembly code associated with each node. We extracted the operator type (e.g. `add`, `mov`, etc.) from each instruction, and then embeds each operator into a vector, the initial node representation is a sum of all operator embeddings.

We searched through the following hyperparameters: (1) triplet or pair training, (2) learning rate in $\{10^{-3}, 10^{-4}\}$, (3) number of propagation layers; (4) share propagation layer parameters or not; (5) GRU vs one-layer MLP for the $f_{\text{node}}$ module.

Overall we found that (1) triplet training performs slightly better than pair training in this case; (2) both learning rates can work but the smaller learning rate is more stable; (3) increasing number of propagation layers generally helps; (4) using different propagation layer parameters perform better than using shared parameters; (5) GRUs are more stable than MLPs and performs overall better.

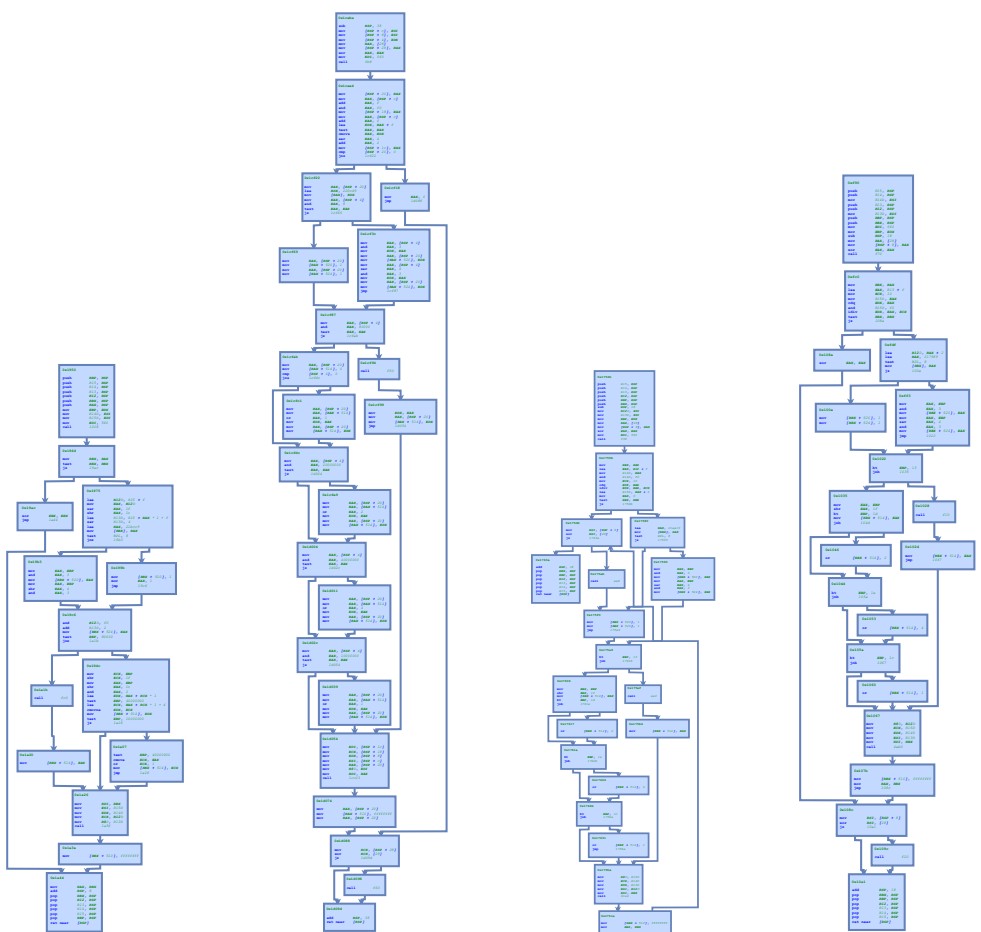

Figure 5: Example control flow graphs for the same binary function, compiled with different compilers (`clang` for the leftmost one, `gcc` for the others) and optimization levels. Note that each node in the graphs also contains a set of assembly instructions which we also take into account when computing similarity using learned features.

In addition to the results reported in the main paper, we have also tried the same models on another dataset obtained by compiling the compression software `unrar` with different compilers and optimization levels. Our graph similarity learning methods also perform very well on the unrar data, but this dataset is a lot smaller, with around 400 functions only, and overfitting is therefore a big problem for any learning based model, so the results on this dataset are not very reliable to draw any conclusions.

A few more control-flow graph examples are shown in Figure 5. The distribution of graph sizes in the training set is shown in Figure 6.

## C    EXTRA ATTENTION VISUALIZATIONS

A few more attention visualizations are included in Figure 7, Figure 8 and Figure 9. Here the graph matching model we used has shared parameters for all the propagation and matching layers and was trained with 5 propagation layers. Therefore we can use a number $T$ different from the number of propagation layers the model is being trained on to test the model's performance. In both visualizations, we unrolled the propagation for up to 9 steps and the model still computes sensible attention maps even with $T > 5$.

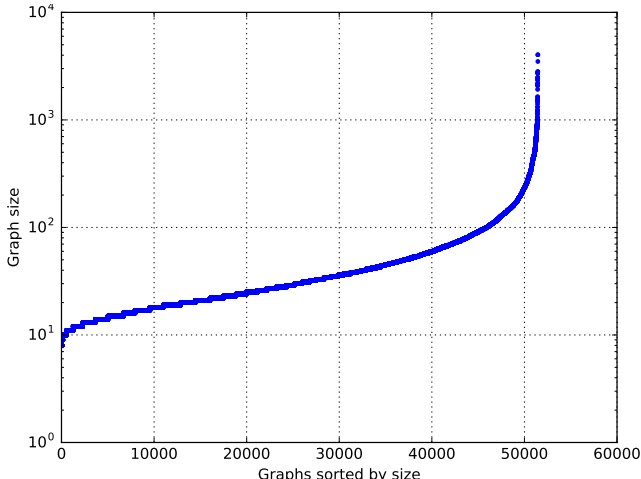

Figure 6: Control flow graph size distribution in the training set. In this plot the graphs are sorted by size on the x axis, each point in the figure corresponds to the size of one graph.

Note that the attention maps do not converge to very peaked distributions. This is partially due to the fact that we used the node state vectors both to carry information through the propagation process, as well as in the attention mechanism as is. This makes it hard for the model to have very peaked attention as the scale of these node state vectors won't be very big. A better solution is to compute separate key, query and value vectors for each node as done in the tensor2tensor self-attention formulation Vaswani et al. (2017), which may further improve the performance of the matching model.

Figure 7 shows another possibility where the attention maps do not converge to very peaked distributions because of in-graph symmetries. Such symmetries are very typical in graphs. In this case even though the attention maps are not peaked, the cross graph communication vectors $\mu$ are still zero, and the two graphs will still have identical representation vectors.

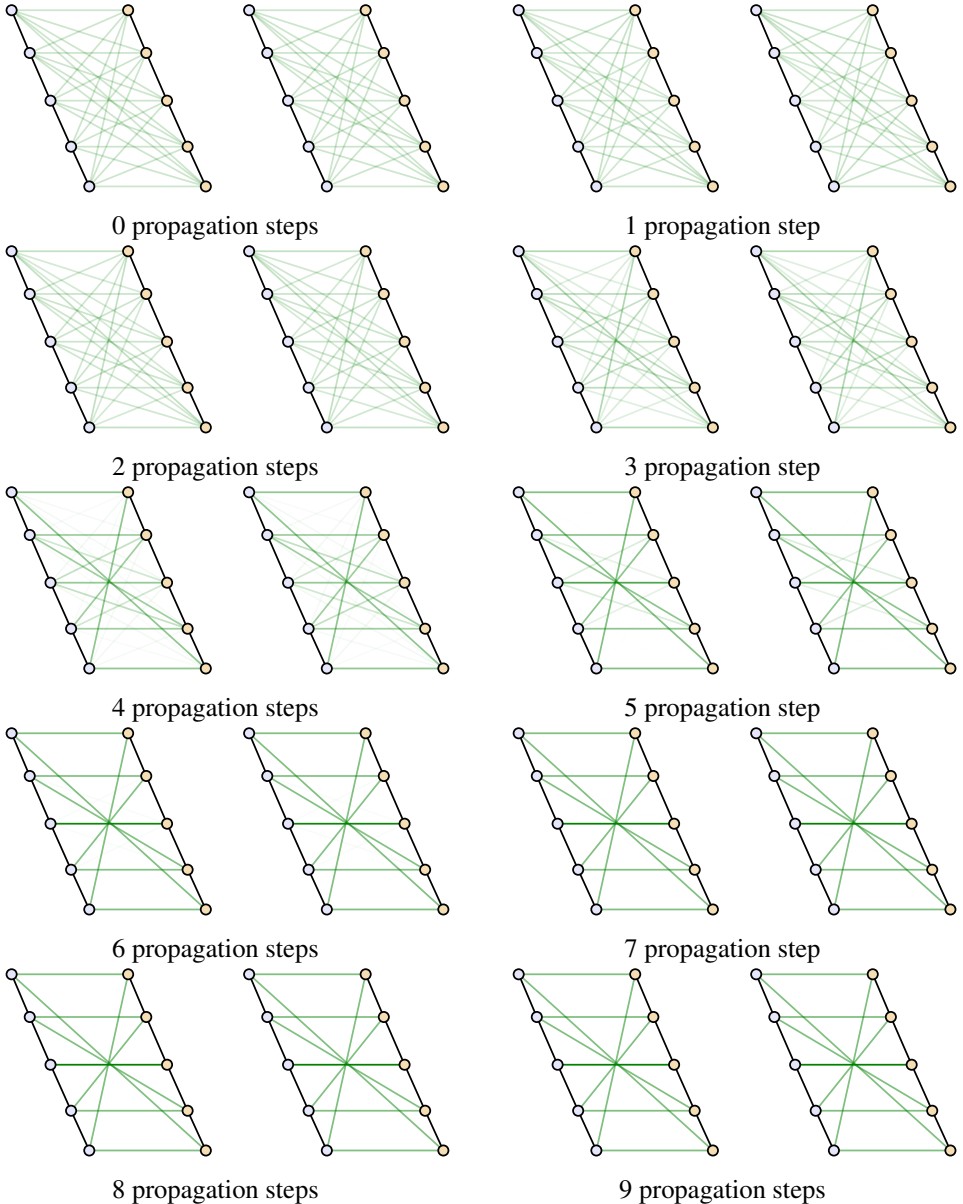

Figure 7: The change of cross-graph attention over propagation layers. Here the two graphs are two isomorphic chains and there are some in-graph symmetries. Note that in the end the nodes are matched to two corresponding nodes with equal weight, except the one at the center of the chain which can only match to a single other node.

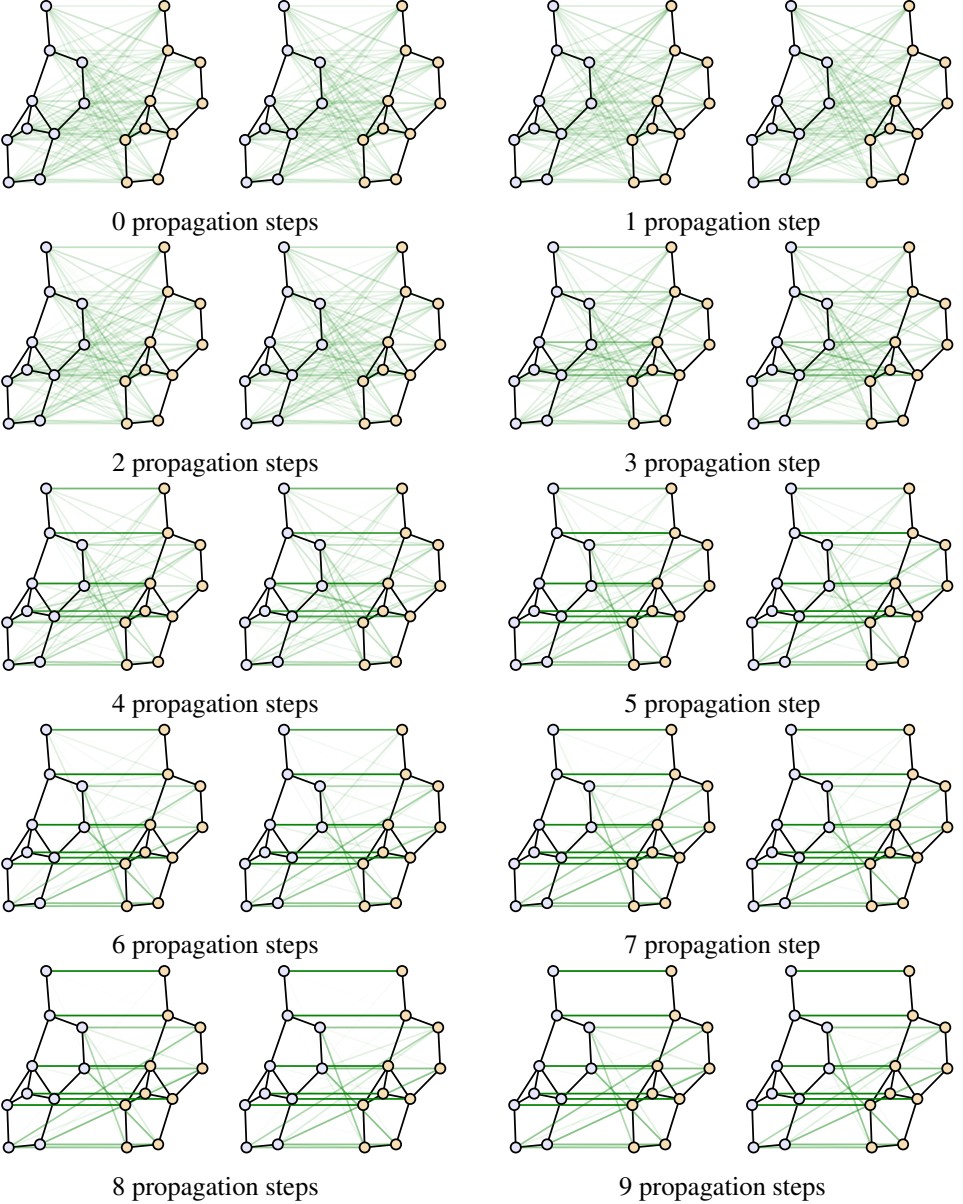

0 propagation steps         1 propagation step

2 propagation steps         3 propagation step

4 propagation steps         5 propagation step

6 propagation steps         7 propagation step

8 propagation steps         9 propagation steps

Figure 8: The change of cross-graph attention over propagation layers. Here the two graphs are isomorphic, with graph edit distance 0. Note that in the end a lot of the matchings concentrated on the correct match.

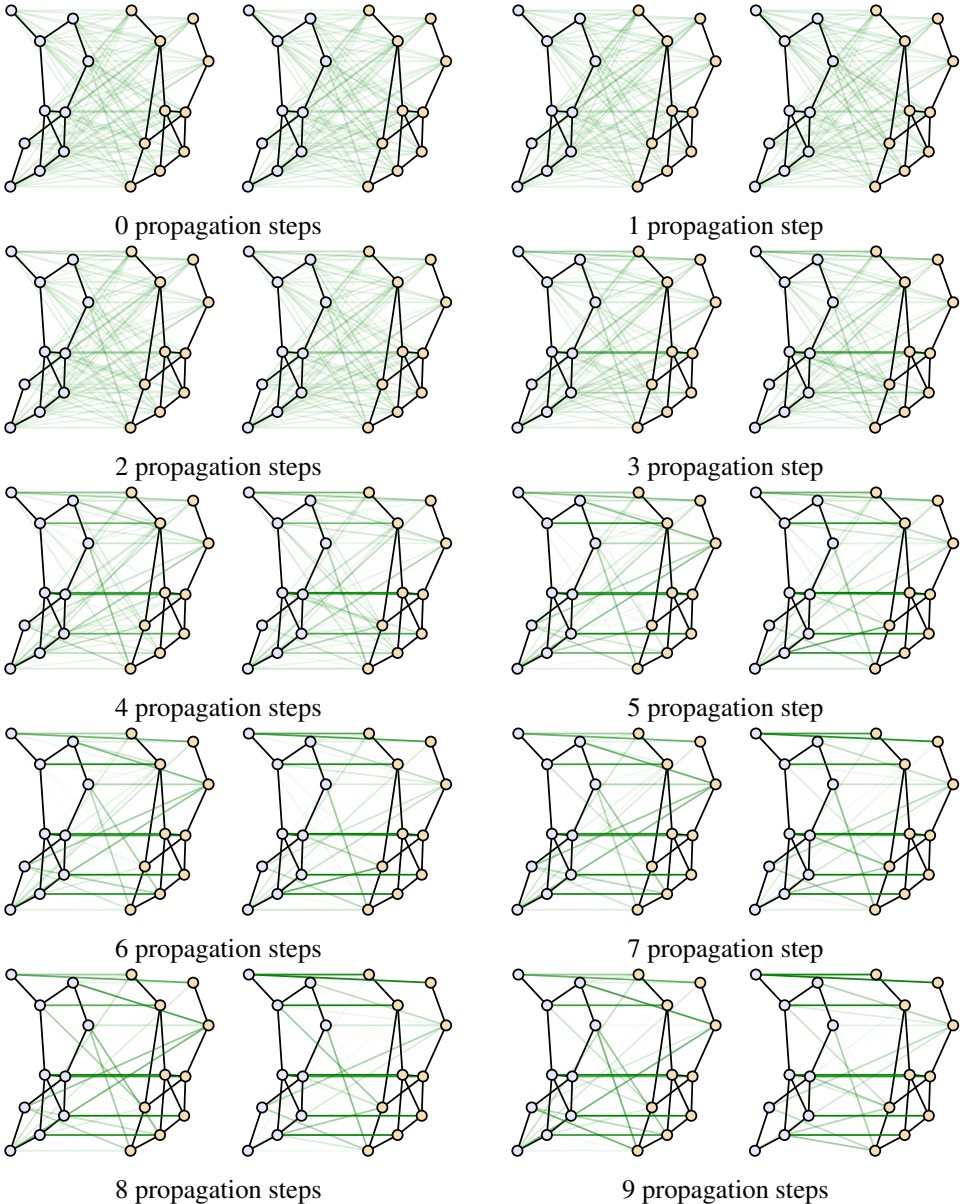

Figure 9: The change of cross-graph attention over propagation layers. The edit distance between these two graphs is 1.

