# OpenReview forum: "Graph Matching Networks for Learning the Similarity of Graph Structured Objects"
_ICLR.cc/2019/Conference_

### Official Review · AnonReviewer1 · 2018-10-29
**Novel concept of cross-graph attention -- lack of in depth discussion**

**Rating:** 6
**Confidence:** 4

**Review:**

The authors present two methods for learning a similarity score between pairs of graphs. They first is to use a shared GNN for each graph to produce independent graph embeddings on which a similarity score is computed. The authors improve this model using pairs of graphs as input and utilizing a cross-graph attention-mechanism in combination with graph convolution. The proposed approach is evaluated on synthetic and real world tasks. It is clearly shown that the proposed approach of cross-graph attention is useful for the given task (at the cost of extra computation).

A main contribution of the article is that ideas from graph matching are introduced to graph neural networks and it is clearly shown that this is beneficial. However, in my opinion the intuition, effect and limitations of the cross-graph attention mechanism should be described in more detail. I like the visualizations of the cross-graph attention, which gives the impression that the process converges to a bijection between the nodes. However, this is not the case for graphs with symmetries (automorphisms); consider, e.g., two star graphs. A discussion of such examples would be helpful and would make the concept of cross-graph attention clearer.

The experimental comparison is largely convincing. However, the proposed approach is motivated by graph matching and a connection to the graph edit distance is implied. However, in the experimental comparison graph kernels are used as baseline. I would like to suggest to also use a simple heuristics for the graph edit distance as a baseline (Riesen, Bunke. Approximate graph edit distance computation by means of bipartite graph matching. Image and Vision Computing, 27(7), 2009).


There are several other questions that have not been sufficiently addressed in the article.

* In Eq. 3, self-attention is used to compute graph level representations to "only focus on important nodes in the graph". How can this be reconciled with the idea of measuring similarities across the whole graph? Can you give more insights in how the attention coefficients vary for positive as well as negative examples? How much does the self-attention affects the performance of the model in contrast to mean or sum aggregation?
* Why do you chose the cross-graph similarity to be non-trainable? Might there be any benefits in doing so?
* The note on page 5 is misleading because two isomorphic graphs will lead to identical representations even if communication is not reduced to zero vectors (this happens neither theoretically nor in practice).
* Although theoretical complexity of the proposed approach is mentioned, how much slower is the proposed approach in practice? As similarity is computed for every pair of nodes across two graphs, the proposed approach, as you said, will not scale. In practice, how would one solve this problem given two very large graphs which do not fit into GPU memory? To what extent can sampling strategies be used (e.g., from GraphSAGE)? Some discussion on this would be very fruitful.


In summary, I think that this is an interesting article, which can be accepted for ICLR provided that the cross-graph attention mechanism is discussed in more detail.


Minor remarks:

* p3: The references provided for the graph edit distance in fact consider the (more specific) maximum common subgraph problem.

---

> ### Author Response · Authors · 2018-11-16
> **Author Response**
>
> > “I think that this is an interesting article, which can be accepted for ICLR provided that the cross-graph attention mechanism is discussed in more detail.”
>
> Thank you for your appreciation.  We hope the comments below can address your concerns.
>
>
>
> > “this is not the case for graphs with symmetries (automorphisms) … A discussion of such examples would be helpful and would make the concept of cross-graph attention clearer.”
>
> Indeed the visualizations are only used to qualitatively show what the models have learned, and in practice it is not easy to see all nodes in the pair of graphs match exactly as a bijection, precisely because of the existence of a lot of symmetry.  This can be demonstrated on some very simple graphs, for example two chains of 5 nodes A1-B1-C1-D1-E1 and A2-B2-C2-D2-E2, we observe that using a trained model A1 is matched to both A2 and E2 because of symmetry, but C1 is only matched to C2.  We have added such a visualization in the revised paper (page 18, figure 7) with a bit more discussion on this.  We will make sure this clear.
>
>
> > “the proposed approach is motivated by graph matching and a connection to the graph edit distance is implied”
>
> The motivation of our graph matching networks model comes from the intuition of more efficiently fusing information from the pair of graphs.  Despite the name, this model is not directly connected to the classic graph matching problems, and the nodes are not explicitly encouraged to match.  The model can choose not to use this cross-graph matching mechanism if it learns not to.
>
> On the other side, we didn’t aim to solve the graph edit distance problem, but rather used it to study the properties of our models, and demonstrate that (1) the graph similarity learning models are competitive against hand-designed baselines and (2) they can be adapted to any similarity metric.  Our models are competitive against WL kernel, but we are also not claiming it is the state-the-art.
>
>
> > “Eq.3, self-attention”
>
> This form of aggregation across nodes first appeared in (Li et al. 2015), and we used exactly the same formulation (we will make this clear in the paper).  In all the experiments we have tried, this gated sum outperformed simple sum or mean aggregation by a significant margin.  The choice of this aggregation function is purely based on performance.
>
>
> > “Why do you chose the cross-graph similarity to be non-trainable?”
>
> Could you clarify?  The h_i vectors are learned, hence the attention weights a_{ij} can adapt as h_i and h_j changes.  The similarity metric s_h is fixed to be a pre-specified metric, we didn’t feel there’s really a need to change it as the h_i’s can adapt to this metric.  In hindsight, making s_h a learnable module may even improve performance further.
>
>
> > “The note on page 5 is misleading because two isomorphic graphs will lead to identical representations even if communication is not reduced to zero vectors”
>
> The note on page 5 says when (1) the nodes can perfectly match and (2) the attention weights are peaked, the cross-graph message vector will be reduced to 0, and hence the node representations will continue to be identical in the next round of propagation.  This doesn’t exclude the possibility that when the cross-graph message vector is non-zero, the node representation can still be identical, as in the example you mentioned.  We will make this clear.
>
>
> > “how much slower is the proposed approach in practice? … how would one solve this problem given two very large graphs”
>
> We have benchmarked the efficiency of GMN models against GNNs.  On the graph edit distance learning problem, GMNs take 1-2x as much time as GNNs on graphs from 20 nodes to 200 nodes, taking ~3 seconds to compute similarity for 1000 pairs of graphs of size 20, and ~80 seconds for 1000 pairs of graphs of size 200, when running on CPU with a tensorflow implementation.
>
> As discussed in the paper, the GMN model scales quadratically w.r.t. number of nodes, making it very expensive for large graphs.  Therefore the setting we targeted in this paper is mostly about computing the similarity of pairs of small graphs, which already has a lot of applications.  On the other hand, sampling may be used for deploying GMNs to large graphs, by for example use a constant number of sampled nodes from the other graph to compute the cross-graph vectors.  Serious discussions and solutions for solving this scaling problem is the topic of another paper.

---

### Official Review · AnonReviewer2 · 2018-11-02
**Interesting  application but inadequate experiments**

**Rating:** 6
**Confidence:** 4

**Review:**

The authors introduce a Graph Matching Network for retrieval and matching of graph structured objects. The proposed methods demonstrates improvements compared to baseline methods. However, I have have three main concerns:
1) Unconvining experiments.
	a) Experiments in Sec4.1. The experiments seem not convincing. Firstly, no details of dataset split is given. Secondly, I am suspicious the proposed model is overfitted, although proposed GSL models seem to bring some improvements on the WL kernel method. As shown in Tab.3, performance of GSL models dramatically decreases when adapting to graphs with more nodes or edges. Besides, performance of the proposed GSLs also drops when adapting to different combines of k_p and k_n as pointed in Sec.B.1. However, the baseline WL kernel method demonstrates favourable generalization ability.

	b）Experiments in Sec4.2. Only holding out 10% data into the testing set is not a good experiment setting and easily results in overfitting. The authors are suggested to hold more data out for testing. Besides, I wonder the generalization ability of the proposed model. The authors are suggested to test on the small unrar dataset mentioned in Sec.B.2 with the proposed model trained on the ffmpeg dataset in Sec4.2.

2) Generalization ability. The proposed model seems sensitive to the size and edge density of the graphs. The authors is suggested to add experiments mentioned in (1).

3) Inference time and model size. Although the proposed model seems to achieve increasing improvements with the increasing propagation layers. I wonder the cost of inference time and model size compared to baselines methods.

---

> ### Author Response · Authors · 2018-11-16
> **Author Response**
>
> > Experiments in sec 4.1:
>
> In this experiment we were training on procedurally generated data, rather than a fixed training set.  We set aside a set of generated data and use these to compare all models.  In this setting the models will not overfit in the traditional sense, as the data generator can be queried as many times as needed to get new data for training from the data distribution, and the models will learn to fit this distribution well.
>
> On the other hand, whether the learned model generalizes to out-of-distribution data is a separate question, and this is a challenge for any learning-based approaches.  In the graph edit distance case, generalizing across graphs of drastically different sizes is very challenging.  In the experiments in the appendix, we are training on graphs of 20-50 nodes, and testing on graphs of 100 or 200 nodes.  This is difficult as graphs with 200 nodes have 10x more nodes than graphs with only 20 nodes, and 100x more edges.
>
> We do not claim our approach can beat the state-of-the-art on graph edit distance, and this experiment is only to show that our model can adapt to any definition of similarity.  On the other hand the WL kernel is a non-learning approach, and cannot easily adapt well to other domains when the graphs are feature-rich.  When used on large graphs WL kernel has a unique advantage over our approach as the effective feature size for a graph scales as O(T|V|), with T being the number of steps to run and |V| the number of nodes in the graph.  On the other hand, the models we studied in the paper used a constant graph feature size of 128.  Training larger models with larger graph features, and on graphs of more diverse sizes can improve the performance of our model.
>
>
> > Experiments in Sec 4.2:
>
> Holding out 10% data as the test set is a widely accepted practice in machine learning, for example the widely used ImageNet dataset contains roughly 10% data in the validation and test sets.
>
> We didn’t expect the model learned on ffmpeg to generalize to unrar, as the two softwares are very different and the distribution of the control flow graphs from the two datasets are also very different.  However, we did try this generalization experiment suggested by the reviewer and to our surprise, the GNN and GMN models does generalize quite well across datasets as shown below:
>
> Training on ffmpeg:
> GMN: 98.01 / 97.48 (test on ffmpeg), 97.84 / 97.92 (test on unrar)
> GNN: 95.58 / 95.21 (test on ffmpeg), 94.66 / 96.35 (test on unrar)
> baseline: 90.24 / 89.61 (test on ffmpeg)
>
> Training on unrar:
> GMN: 96.23 / 97.92 (test on unrar), 91.82 / 91.56 (test on ffmpeg)
> GNN: 95.19 / 94.79 (test on unrar), 86.36 / 85.77 (test on ffmpeg)
> baseline: 87.22 / 87.50 (test on unrar)
>
> Here the two numbers in each pair are pair AUC and triplet accuracy.  The models we tested are the ones that uses the graph structure only.
>
> We can see that the models trained on ffmpeg clearly transfers very well to unrar, with performance on par with and in some cases even better than models trained on unrar alone.  While in the opposite direction, models trained on unrar doesn’t transfer as well to ffmpeg as overfitting is more of a problem on the unrar dataset because it is significantly smaller.
>
>
> > Experiments in Sec 4.3:
>
> The f_node in our formulation refers to a module that updates node representations, and we have tested MLPs, RNNs, GRUs and LSTMs for this.  When we say “to aggregate the messages, we use a simple sum” we were referring to the sum inside eq.2, not the form of f_node.  The GNN model described in this paper is very similar to the GCNs but more general. In particular the GNN model we described can use nonlinear MLPs for both message computation and node updates.  See references (Gilmer et al. 2017; Battaglia et al. 2018) for more discussions about different variants of GNN architectures.
>
>
> > Inference time and model size:
>
> In the paper we have discussed the computation complexity of our models (see e.g. Sec 3.2 and Sec 5).  The GNN model scales linearly w.r.t. the number of propagation layers T, the dimensionality of h_i (D), and the number of nodes (|V|) and edges (|E|), as O(TD(|V| + |E|)).  The GMN model on the other hand scales as O(TD|V|^2).
>
> We have benchmarked the efficiency of GMN models against GNNs.  On the graph edit distance problem, GMNs take 1-2x as much time as GNNs on graphs from 20 nodes to 200 nodes, taking ~3 seconds to compute similarity for 1000 pairs of graphs of size 20, and ~80 seconds for 1000 pairs of graphs of size 200, when running on CPU with a tensorflow implementation.
>
> Comparing the wall clock time of different implementations of different approaches optimized to different levels is not very meaningful.  But just to provide an extra data point, the open source  WL kernel implementation (written in python) we used is >10x slower than both the GMN and GNNs on the same 1000 pairs of graphs with the same T, which scales as O(T(|V| + |E|)) in principle.

---

### Official Review · AnonReviewer3 · 2018-11-05
**A good paper, but requires more ablation study**

**Rating:** 5
**Confidence:** 4

**Review:**

Graph matching is a classic and import problem in computer vision, data mining, and other sub-areas of machine learning. Previously, the graph matching problems are often modeled as combinatorial optimization problems, e.g. Quadratic Assignment Problems. While these optimization problems are often NP-hard, researchers often focus on improving the efficiency of the solvers. The authors attack the problem in another way. They proposed an extension of graph embedding networks, which can embed a pair of graphs to a pair of vector representations, then the similarity between two graphs can be computed via computing the similarities of the pair of vector representations. The proposed model is able to match graphs in graph level as it can predict the similarities of the two graphs.

Compare to Graph Embedding Networks (GNN), the authors proposed a new model, in which a new matching module accepts the hidden vector of nodes from two graphs and maps them to a hidden matching variable, then the hidden matching variable serves as messages as in Graph Embedding Networks. This is the main contribution of the paper compared to GNN.

The main problem of the paper is that it is not clear where the performance improvement comes from. The authors proposed a cross-graph attention-based matching module. However, it is not clear whether the performance improvement comes from the cross-graph interaction, or the attention module is also important. It would be nice if the author can do some ablation study on the structure of the new matching module.

In graph matching, we not only care about the overall similarity of two graphs but also are interested in finding the correspondence between the nodes of two graphs, which requires the similarities between vertexes of two graphs. Compared to another [1] deep learning based graph matching model, the author did not show that the proposed are able to give the matching constraints. For example, while the authors show that it is possible to use GMN to learn graph edit distances, is it possible to use the GMN to help to the exact editing?



[1] Zanfir, Andrei, and Cristian Sminchisescu. "Deep Learning of Graph Matching." Proceedings of the IEEE Conference on Computer Vision and Pattern Recognition. 2018.

---

> ### Author Response · Authors · 2018-11-16
> **Author Response**
>
> Q: “it is not clear where the performance improvement comes from…  not clear whether the performance improvement comes from the cross-graph interaction, or the attention module is also important ... nice if the author can do some ablation study on the structure of the new matching module.”
>
> A: The experiments we did in section 4.1 and 4.2 is exactly designed to address two questions: (1) can we learn graph similarity metrics from data, and be competitive with non-learning baselines, and (2) is the cross-graph matching module really useful.  The experiment results clearly showed both.
>
> In particular, to isolate the contributions from different factors, we made sure that when comparing the graph embedding model (GNN) and the graph matching model (GMN), all the possible confounding factors are carefully controlled*, and the only difference between them is that GMNs use the cross-graph matching mechanism, while GNNs do not.  Therefore the results reported in section 4.1 and 4.2 can clearly show that the cross-graph matching mechanism really does provide a significant improvement.
>
> In addition to this, we provided more ablation studies in section 4.3, where we compared the effect of different node update modules f_node (GNN vs GCN), and different strategies for fusing information from both graphs (embedding vs Siamese vs matching).  We believe this is a reasonably thorough study of these important modeling decisions, and again the results show that the improvement we get from using cross-graph-matching is consistent.
>
>
> Q: “... we not only care about the overall similarity of two graphs but also are interested in finding the correspondence between the nodes of two graphs, which requires the similarities between vertexes of two graphs ...”
>
> A: Graph matching is an important problem, but it is not the problem we address in this paper.  The goal of the matching problem tackled in [1] is to find accurate matching between two sets of vertices, while the problem we are solving in this paper is mostly focused on learning graph-level similarity metrics.  Such a problem alone has a lot of practical applications.  On the other side, as the reviewer commented, we are also interested in finding correspondence between the nodes.  Since the focus of the approaches described in this paper is to estimate the similarity between graphs, the models don’t have any explicit incentive to really match nodes.  However, we did show through the visualization of the attention maps that our GMN model can find some soft node-level matches, which is a nice by-product of our approach.
>
>
> * We did thorough hyperparameter searches over all the important hyperparameters, including number of propagation layers T, dimensionality of h_i, the type of the aggregation function (eq.3), learning rate etc., and the set of candidate hyperparameter combinations are the same for all models.

---

### Meta-Review · Area_Chair1 · 2018-12-20
**Very borderline paper**

**Confidence:** 4
**Recommendation:** Reject

**Metareview:**

This is a tough choice as it is a reasonably strong paper.
I am similar to another reviewer quite confused how this graph matching can "only focus on important nodes in the graph"
This seems counter-intuitive and the only reason given in the rebuttal is that other people have done it also..

Relatedly: "In graph matching, we not only care about the overall similarity of two graphs but also are interested in finding the correspondence between the nodes of two graphs"

I am sorry for the authors and hope they will get it accepted at the next conference.